# Shield Decentralization for Safe Multi-Agent Reinforcement Learning

**Daniel Melcer**
Northeastern University
Boston, MA 02115
`melcer.d@northeastern.edu`

**Christopher Amato**[*]
Northeastern University
Boston, MA 02115
`c.amato@northeastern.edu`

**Stavros Tripakis**[*]
Northeastern University
Boston, MA 02115
`stavros@northeastern.edu`

## Abstract

Learning safe solutions is an important but challenging problem in multi-agent reinforcement learning (MARL). Shielded reinforcement learning is one approach for preventing agents from choosing unsafe actions. Current shielded reinforcement learning methods for MARL make strong assumptions about communication and full observability. In this work, we extend the formalization of the shielded reinforcement learning problem to a decentralized multi-agent setting. We then present an algorithm for decomposition of a centralized shield, allowing shields to be used in such decentralized, communication-free environments. Our results show that agents equipped with decentralized shields perform comparably to agents with centralized shields in several tasks, allowing shielding to be used in environments with decentralized training and execution for the first time.

## 1 Introduction

Recently, the advent of deep reinforcement learning has produced solutions in highly complex domains such as Atari [17], and the game of Go [20]. However, the neural networks which power these methods are opaque, and there is a significant risk of unintentional harm when a poorly understood reinforcement learning method is applied to a safety-critical system [2].

One common approach for safe RL is the idea of a shield—a reactive system which monitors a reinforcement learning agent and environment [1]. During each step, the agent proposes an action, the shield evaluates whether this action is safe, and then the shield potentially replaces the proposed action with a known-safe action. This method has been extended to multi-agent environments, but the shield remains centralized [10]. Such methods assume instantaneous communication with a centralized coordination algorithm, which is not realistic in many environments.

In this work, we first discuss the centralized shield synthesis problem, and outline its proposed solutions. We describe several implicit communication assumptions present in the existing centralized shield definition, and define a novel framework for decentralized shielding. These decentralized shields enable each agent to act independently, without any coordination after shield synthesis.

---

[*]Equal Advising

36th Conference on Neural Information Processing Systems (NeurIPS 2022).

Our main technical contribution is an algorithm which, using any centralized shield as input, produces a decentralized shield that guarantees safety. We also discuss various properties of input centralized shields, and the effect of these properties on the permissiveness of the output shield.

Lastly, we evaluate the performance of our shield decomposition algorithm on shields from several domains. We first analyze the structure of the shield itself, ensuring that the majority of actions allowed by the centralized shield are still allowed by the decentralized shield. We then train a shielded reinforcement learning agent to solve these tasks, and measure the performance of the learned policy. We find that all agents achieve a comparable value in the environment, and that, as is to be expected from the theoretical guarantees of our framework, decentralized shielding achieves 100% safety. These results show that our shield decentralization approach can, for the first time, allow reinforcement learning methods to be provably safe in decentralized multi-agent settings.

## 2 Related Work

The concept of shielded reinforcement learning [1] draws from the idea of Runtime Enforcement [6], from the Formal Methods community. In the single-agent case, shielding has been extended to environments with continuous action spaces [8], stochastic or adversarial environments [15], and partially-observable environments [7]. As with all shielding methods, these extensions require a model of the environment, with a varying degree of abstraction. This differs slightly from contract-based decomposition [5]; instead of defining a contract first, and then constructing systems whose interfaces satisfy this contract, shielding involves constraining an agent to act safely within an environment whose dynamics are fixed.

Other proposed methods specify algorithms for enforcing safety in the presence of communication constraints or delays for pathfinding agents [23], or for general LTL specifications [12], but neither method eliminates communication entirely.

Several pieces of work focus on verifying a neural network or closed-loop reinforcement learning system after training is complete [4, 14]. These methods could potentially succeed in a multi-agent reinforcement learning system if successfully scaled up, but such approaches typically make no guarantees of safety during training.

## 3 Preliminaries

We first describe several relevant structures, and a formalization of the centralized shielding problem. Many elements of this framework have been described in prior formulations [1, 6], and extended to the multi-agent setting [10].

Let $\Delta(X)$ be the set of all probability distributions over set $X$. Let $\mathbb{R}$ denote the set of reals.

An environment with cooperative agents and full observability is often characterized as a *MMDP*:

**Definition 1 (MMDP)** *A* Multi-agent Markov Decision Process (MMDP) *is a tuple* $\mathcal{M} = (\mathbb{D}, \mathbb{S}, \mathbb{A}, \mathcal{E}, T, \gamma, R, b_0)$ *where* $\mathbb{D} = \{1, \ldots, n\}$ *is a set of agents,* $\mathbb{S}$ *is a set of states,* $\mathbb{A} = \prod_{i \in \mathbb{D}} \mathbb{A}_i$ *is a finite joint action space, factorizable into* $n$ *individual action spaces,* $\mathcal{E} : \mathbb{S} \to \mathcal{P}(\mathbb{A})$ *represents the set of actions available to the agents at a given state,* $T : \mathbb{S} \times \mathbb{A} \to \Delta(\mathbb{S})$ *is the transition probability distribution function,* $\gamma \in [0, 1]$ *represents the discount factor for future rewards,* $R : \mathbb{S} \times \mathbb{A} \to \mathbb{R}$ *is the reward given after a transition, and* $b_0 \in \Delta(\mathbb{S})$ *is the distribution of initial states.*

We call a sequence of states and joint actions $((s_0, s_1, \ldots, s_{n+1}), (a_0, a_1, \ldots, a_n))$ an *environment trace* of MMDP $\mathcal{M}$ if every pair $(s_i, a_i, s_{i+1})$ is allowed by the transition function of the MMDP.

$\mathcal{M}$ is said to be in a *deadlock* at state $s \in \mathbb{S}$ if $\mathcal{E}(s) = \emptyset$. $\mathcal{M}$ is *deadlock-free* if there are no environment traces of $\mathcal{M}$ that end in a deadlock state.

We will often have some prior knowledge of how to abstract the states in a meaningful way for safety properties. For example, in a domain where we avoid collisions between agents, the relative positions of the agents may be useful, even if the absolute positions of all agents are not known. We call a function $f : \mathbb{S} \to L$ which translates MMDP states into members of a given *label set* an *abstraction function*. Note that some prior work refers to $f$ as the *observer function* [1, 10]; we

use a different name to avoid potentially conflicting terminology. Every environment trace through a MMDP has a corresponding *label trace* $((l_0, a_0), \ldots, (l_n, a_n))$, where $\forall i \in \{0, \ldots, n\}, l_i = f(s_i)$.[1] Several unique environment traces may correspond to the same label trace.

We note that an environment with partial observability or a more complex reward structure is acceptable. As long as each agent can independently calculate the current label, a shield can be constructed using this framework. However, for simplicity, we continue to use MMDPs in our description.

**Definition 2 (DFA)** *A Deterministic Finite Automaton (DFA) is a tuple $\phi = (Q, \Sigma, \delta, s_0, F)$ where $Q$ is a set of states, $s_0 \in Q$ is the initial state, $\Sigma$ is an alphabet, $\delta : Q \times \Sigma \to Q$ is the transition function, and $F \subseteq Q$ is the set of accepting states.*

Given a word $(w_0, w_1, \ldots, w_n) \in \Sigma^*$, the corresponding *DFA trace* $(q_0, q_1, \ldots, q_{n+1}) \in Q^*$ is obtained by stepping through the transition function. A word is *accepted* by the DFA iff the last state of its corresponding DFA trace is a member of $F$.

For a given MMDP $\mathcal{M} = (\mathbb{D}, \mathbb{S}, \mathbb{A}, \mathcal{E}, T, \gamma, R, b_0)$ and label set $L$, we define a *safety specification* over $\mathcal{M}$ and $L$ to be a DFA $\phi^s = (Q^s, (L \times \mathbb{A}), \delta^s, s_0^s, F^s)$ which accepts the empty word and does not contain any transitions from $Q^s \setminus F^s$ to $F^s$. $\mathcal{M}$ satisfies $\phi^s$ ($\mathcal{M} \vDash \phi^s$) iff all possible label traces of $\mathcal{M}$ are accepted by $\phi^s$. $\phi^s$ induces the *accepting action function* $\mathcal{C}^{\phi^s} : Q^s \times L \to \mathcal{P}(\mathbb{A}) = (q, l) \to \{a \in \mathbb{A} | \delta^s(q, (l, a)) \in F^s\}$; given a DFA state and label, output the actions which transition the DFA to an accepting state.

We can compose a DFA $\phi^g$ and a MMDP $\mathcal{M}$ to form a new MMDP $(\mathcal{M} || \phi^g)$. The state space of the resulting MMDP is the product of the state spaces of $\mathcal{M}$ and $\phi^g$, and the transition function steps through the DFA and MMDP's states in lockstep. At state $(s, q)$, a given action is only enabled in the resulting MMDP if it is a member of both $\mathcal{E}(s)$ and $\mathcal{C}^{\phi^g}(q, f(s))$.

# 4    Centralized Shielding

We use a definition of a centralized shield based on the joint actions which are allowed at each shield state. Note that some prior formulations define the shield as a Moore machine which outputs a specific safe action in response to an input action [6]; the following definition is equivalent.

**Definition 3 (Centralized Shield)** *A DFA $\phi^g$ is a* centralized shield *for MMDP $\mathcal{M}$ that enforces safety specification $\phi^s$ iff $\mathcal{M} || \phi^g \vDash \phi^s$ and $\mathcal{M} || \phi^g$ is deadlock-free.*

**Problem 1 (Centralized Shield Synthesis)** *Given a MMDP $\mathcal{M} = (\mathbb{D}, \mathbb{S}, \mathbb{A}, \mathcal{E}, T, \gamma, R, b_0)$, a label set $L$, an abstraction function $f$ over $\mathcal{M}$ and $L$, and a safety specification $\phi^s$ over $\mathcal{M}$ and $L$, synthesize a centralized shield for $\mathcal{M}$ that enforces $\phi^s$.*

This problem has previously been explored in [1] for the single-agent case, and [10] for the multi-agent case; we do not aim to advance the state of the art in this domain. Briefly, centralized shield synthesis works by constructing a game with two players: the agents, and the environment. When constructing a shield for a multiagent environment, all agents are collectively considered one "player" whose action set equals the space of joint actions for all of the agents. The environment proposes a label, and the agents must respond with a joint action which does not violate the safety specification. The safety game is solved, typically by iterating to a fixpoint [9].

A complete model of the environment is not necessary for this synthesis; an abstraction which captures all possible label traces is sufficient. For example, our test environments all involve avoiding collisions. We therefore use the relative positions of the agents as the label set. The environment abstraction only captures information about how the relative positions of the agents change with each joint action, not any information about rewards or walls.

When using a centralized shield, the agents never explicitly construct the full composition of $\mathcal{M} || \phi^g$, as $\mathcal{M}$ may have an infinite state space, and its exact dynamics may not be known. Instead, the agents maintain the current centralized shield state $g$, initialized to $s_0^g$. At every time step, they observe the label $l$ from the environment. Using the shield's induced accepting action function, the agents compute the set of safe actions $\mathcal{C}^{\phi^g}(g, l)$.

---

[1]We intentionally omit $f(s_{n+1})$ to allow label traces to be a word of $(L \times \mathbb{A})^*$.

At this point, there are two common design choices in how the shield gets used [1]. In *pre-posed shielding*, the agent's action space is restricted to $\mathcal{C}^{\phi^g}(g, l)$; it is unable to choose an unsafe action. In *post-posed shielding*, the agent is free to choose any action. However, if the action is not safe, the shield substitutes the action with an arbitrary action $a' \in \mathcal{C}^{\phi^g}(g, l)$. The agent observes, and is trained on the environment transition $(s, a', r, s')$. The agent is additionally trained on the synthetic transition $(s, a, r + r_p, s')$ where $r_p$ is a penalty reward. For both designs, the shield's transition function $\delta^g$ is then used to obtain the next centralized shield state. We choose to use post-posed shielding for our experiments, as this method is generalizable to a greater variety of agents.

## 5   Decentralized Shielding

The use of a centralized shield requires agents to communicate with each other for two reasons:

First, the set of safe individual actions for a given agent may depend on the action choice of all other agents. For example, consider the safety specification where an even number of agents must pick a given action. Unless a single joint action is decided upon in advance, the agents must all communicate before action selection in order to choose a safe action.

Furthermore, even if the set of safe individual actions are independent, the centralized shield may transition to different states in response to different joint actions—even if both joint actions produce the same result in the environment. Alshiekh et al. [1] gives the example of an industrial valve controller, where the flow of water through the valve is stochastic when the valve is open. Consider a similar environment where multiple agents each control an independent valve, and are unable to observe other agents' valve states, but they do observe the flow of water through each valve. An agent would not be able to tell the difference between other agents' valves being closed, versus their valves being open without water flowing during a given step due to chance. Depending on the shield structure, there may be no way to calculate what the next shield state is based on flow rates alone.

Therefore, in a no-communication environment, each agent must be equipped with a shield which has the following two properties: First, each agent must be able to independently choose actions. No matter what combination of actions are chosen by the various agents, the joint action must be safe. We call a shield with this property *Cartesian*, as the safe joint action set results from the Cartesian product of each agents' safe individual action sets. Second, each agent's local shield must be able to independently determine its next internal state, without direct knowledge of the actions that other agents have taken. We call a shield with this property *unambiguous*.

We call a DFA an *individual shield* for agent $i \in \mathbb{D}$ if the transition alphabet of this DFA is $L \times \mathbb{A}_i$. If the product[2] of a series of individual shields forms a centralized shield which enforces specification $\phi^s$, we say that the individual shields collectively form a *decentralized shield* which enforces $\phi^s$.

**Problem 2 (Dec-A Shield Synthesis)**  *Given a MMDP $\mathcal{M} = (\mathbb{D}, \mathbb{S}, \mathbb{A}, \mathcal{E}, T, \gamma, R, b_0)$, a label set $L$, an abstraction function over $\mathcal{M}$ and $L$, and a safety specification $\phi^s$, synthesize a decentralized shield which enforces $\phi^s$.*

Problem 2 is generally hard to solve, and may even be undecidable in some cases, as it is related to decentralized controller synthesis problems [18, 21]. Therefore, instead of solving Problem 2 directly, we opt for the following approach. First, we attempt to synthesize a centralized shield (i.e., to solve Problem 1) using the methods described in Section 4. If this fails, i.e., if no centralized shield exists, than we can conclude that no decentralized shield exists either. If, on the other hand, we manage to synthesize a centralized shield, than we can *decentralize* it, thereby turning our decentralized shield synthesis problem into the following *shield decomposition* problem:

**Problem 3 (Dec-A Shield Decomposition)**  *Given a MMDP $\mathcal{M} = (\mathbb{D}, \mathbb{S}, \mathbb{A}, \mathcal{E}, T, \gamma, R, b_0)$ and a centralized shield $\phi^g$ for $\mathcal{M}$, synthesize a decentralized shield for $\mathcal{M}$ which enforces $\phi^g$.*

Problem 3 admits a trivial solution, where for every shield state and label, we prescribe a single safe joint action in advance. Such a joint action is guaranteed to exist for all reachable states due to the deadlock-free property of the centralized shield. While the result is technically a decentralized

---

[2]Unlike a standard automaton product, we synchronize on $L$, so the resulting automaton's alphabet is $L \times \mathbb{A}$.

shield that enforces $\phi^s$, this trivial solution does not allow the agents any freedom to explore actions and optimize for expected returns.

Instead, we would ideally like to synthesize a *maximally permissive* shield; i.e. a shield for which there does not exist an alternate shield that admits a strict superset of label traces. However, this is a fundamentally difficult problem in the general case. We therefore present a procedure which produces a maximally permissive decentralized shield when the input centralized shield is Cartesian and unambiguous, and in all other cases produces a shield which is at least as permissive as the trivial solution. The steps of our algorithm are illustrated in Figure 1, and described in what follows.

---

**Algorithm 1** Step 1 of Shield Decomposition: Determining Safe Actions

---

1: **Input**
2: $\quad \phi^g = (G, (L \times \mathbb{A}), \delta^g, s_0^g, F^g)$ $\qquad\qquad$ // A centralized shield where $\mathbb{A} = \prod_{i \in \mathbb{D}} \mathbb{A}_i$
3: **Output**
4: $\quad \mathcal{D}_i : G \times L \to \mathcal{P}(\mathbb{A}_i)$ $\qquad\qquad$ // A set of safe individual actions for agent $i$
5: $\quad \mathcal{R} : G \times L \to [L \to G]$ $\qquad$ // Uniquely determine the next state, given the observed label
6: **procedure** CALCULATESAFEACTIONS($\phi^g$)
7: $\quad$ **for** $s \in G, l \in \mathcal{L}(s)$ **do** $\quad$ // $\mathcal{L}(s)$ = set of all labels which may be observed in shield state $s$
8: $\qquad \mathcal{C}(s,l) \leftarrow \{a \in \mathbb{A} | \delta^g(s,(l,a)) \in F^g\}$ $\qquad\qquad$ // All legal joint actions from $s, l$
9: $\qquad$ Pick arbitrary $a = (a_1, a_2, \ldots, a_n) \in \mathcal{C}(s,l)$ // Guaranteed to exist, given that $l \in \mathcal{L}(s)$
10: $\qquad \forall i \in \mathbb{D} : \mathcal{D}_i(s,l) = \{a_i\}$
11: $\qquad \mathcal{R}(s,l) \leftarrow [\mathcal{L}(\delta^g((s,l),a)) \to \delta^g((s,l),a)]$
12: $\qquad$ **for** $i \in \mathbb{D}$ **do** $\qquad\qquad$ // Optimization: randomly permute $\mathbb{D}$ at each time step
13: $\qquad\quad$ **for** $a_i' \in \mathbb{A}_i$ **do**
14: $\qquad\qquad A \leftarrow \{a_i'\} \times \prod_{j \in \mathbb{D}, j \neq i} \mathcal{D}_j(s,l)$ $\qquad$ // New joint actions if $a_i'$ were added to $\mathcal{D}_i$
15: $\qquad\qquad$ **if** $A \subseteq \mathcal{C}(s,l)$ **and** UNAMBIGUOUSACTIONS($\mathcal{R}(s,l), s, l, A$) **then**
16: $\qquad\qquad\quad \mathcal{D}_i(s,l) \leftarrow \mathcal{D}_i(s,l) \cup \{a_i'\}$
17: $\qquad\qquad\quad \forall a \in A, \mathcal{R}(s,l) \leftarrow \mathcal{R}(s,l) \cup [\mathcal{L}(\delta^g(s,(l,a))) \to \delta^g(s,(l,a))]$
18: $\qquad\qquad$ **end if**
19: $\qquad\quad$ **end for**
20: $\qquad$ **end for**
21: $\quad$ **end for**
22: $\quad$ **return** $\mathcal{D}_i \forall i \in \mathbb{D}, \mathcal{R}$
23: **end procedure**

---

First, for every centralized shield state, and for every label which can be observed from that state, find a subset of safe joint actions which is Cartesian and unambiguous. This step works by first choosing a single safe joint action, like in the trivial solution. The algorithm then attempts to add individual actions for each agent in turn. When considering a given individual action, the algorithm first calculates the set of all joint actions which would be enabled if this individual action were enabled. For example, if the algorithm has previously decided that agent 1 may take either actions $a$ or $b$, then enabling agent 2's action $b$ would require checking that both joint action candidates $(a,b)$ and $(b,b)$ are suitable. In this manner, the set of enabled joint actions remains Cartesian. If any joint action candidate is unsafe, or if adding any of them would cause ambiguity in the next state, then the individual action is rejected. Since the starting joint action and the order of candidate individual actions are arbitrary, this algorithm's output is not unique. The details of this step are further illustrated in Figure 2, and described more specifically in Algorithm 1.

Algorithm 1 performs most of the work in this sequence of algorithms; it takes a centralized shield as input, and outputs a set of safe actions for every state-label combination, as well as the information necessary to determine the next state of the environment.

Algorithm 2 (in appendix) projects the actions found by the previous step into an input-output state machine for each agent, which we call a *transient-state individual shield*. This structure has two categories of states: label states, which receive an observation from the environment (shown as black dots in Figure 1); and action states, which show the allowed actions of the agent (shown as white circles). Lastly, Algorithm 3 (in appendix) performs some mild post-processing so that this structure conforms to the shield interface. Additional information about these structures and algorithms can be found in Section A.

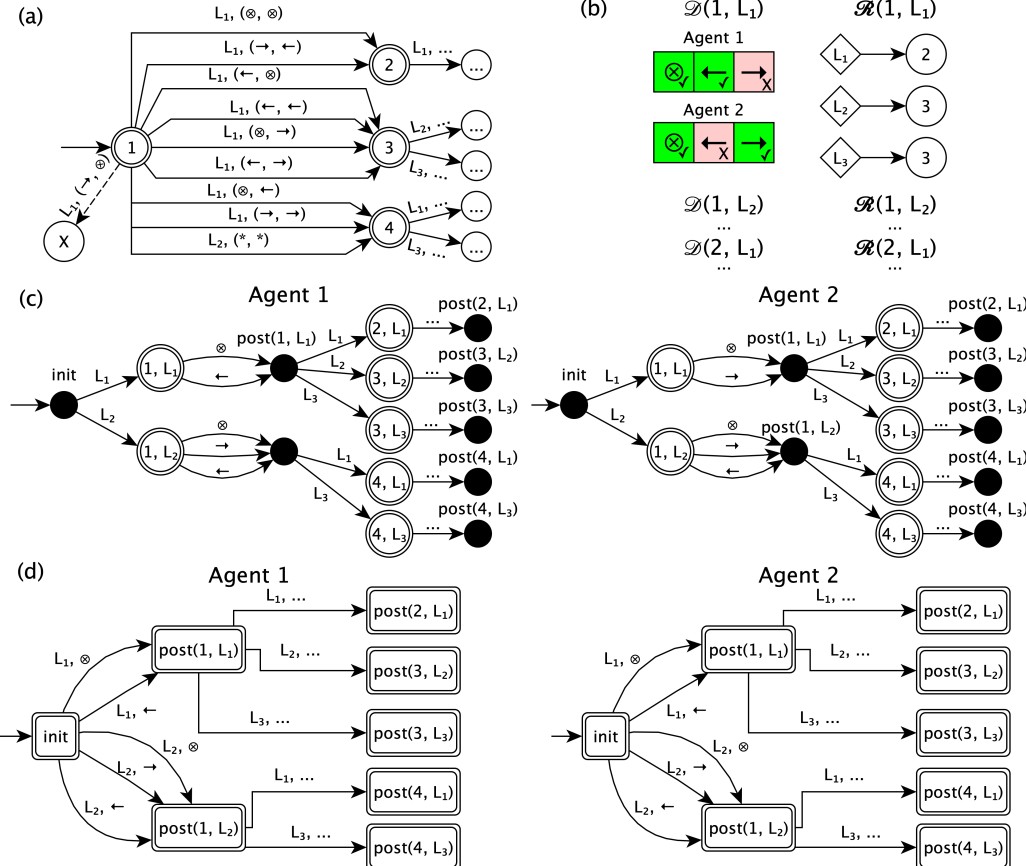

Figure 1: An overview of our algorithm sequence, in a hypothetical environment where $\mathbb{D} = \{1, 2\}, \mathbb{A}_1 = \mathbb{A}_2 = \{\otimes, \leftarrow, \rightarrow\}, \{L_1, L_2, L_3\} \in L$. (a) The input centralized shield, showing $\mathcal{L}(1) = \{L_1, L_2\}, \mathcal{L}(2) = \{L_1\}, \mathcal{L}(3) = \{L_2, L_3\}, \mathcal{L}(4) = \{L_1, L_3\}$, as well as several transitions. (b) The output of Algorithm 1: $\mathcal{D}_{1..n}$, which denotes the allowed actions for each agent, and $\mathcal{R}$, which maps labels seen in the next state to the state number. Here, for shield state 1 and label $L_1$, it is safe for agent 1 to take actions $\otimes$ or $\leftarrow$, while agent 2 may take $\otimes$ or $\rightarrow$. (c) The output of Algorithm 2: transient-state individual shields for each agent, constructed using $\mathcal{D}_{1..n}$ and $\mathcal{R}$. (d) The output of Algorithm 3: individual shields, constructed by transforming the transient-state individual shields. These can be used by agents in communication-free environments to ensure safety.

If the centralized shield given as input to this sequence of algorithms enforces a given specification, the decentralized shield will also enforce that specification. This important safety property is formalized as the following theorem, and proven in Section B.2 of the appendix:

**Theorem 1** *Given a MMDP $\mathcal{M} = (\mathbb{D}, \mathbb{S}, \mathbb{A}, \mathcal{E}, T, \gamma, R, b_0)$ where $\forall s \in \mathbb{S}, \mathcal{E}(s) = \mathbb{A}$, and a centralized shield $\phi^g$ for $\mathcal{M}$, the result of Algorithms 1, 2, and 3 is a decentralized shield for $\mathcal{M}$ which enforces $\phi^g$.*

As mentioned previously, given an input centralized shield which is already both unambiguous and Cartesian, the resulting decentralized shield will be maximally permissive. A shield is trivially unambiguous if each label only appears in a single state; many safety specifications, especially in fully observable environments, are enforceable by such a shield. This includes all *stateless* shields—shields where $F^g = \{s_0^g\}$. For input shields which are non-Cartesian, our algorithm gives one agent "right of way"; an agent with higher priority cannot put another agent in a situation for which there are no safe actions, but the lower-priority agent must otherwise accommodate any potential actions that the higher-priority agent may take. A centralized shield is usually Cartesian in states where such a "right of way" is not necessary, i.e. when agents are not in contention with each other.

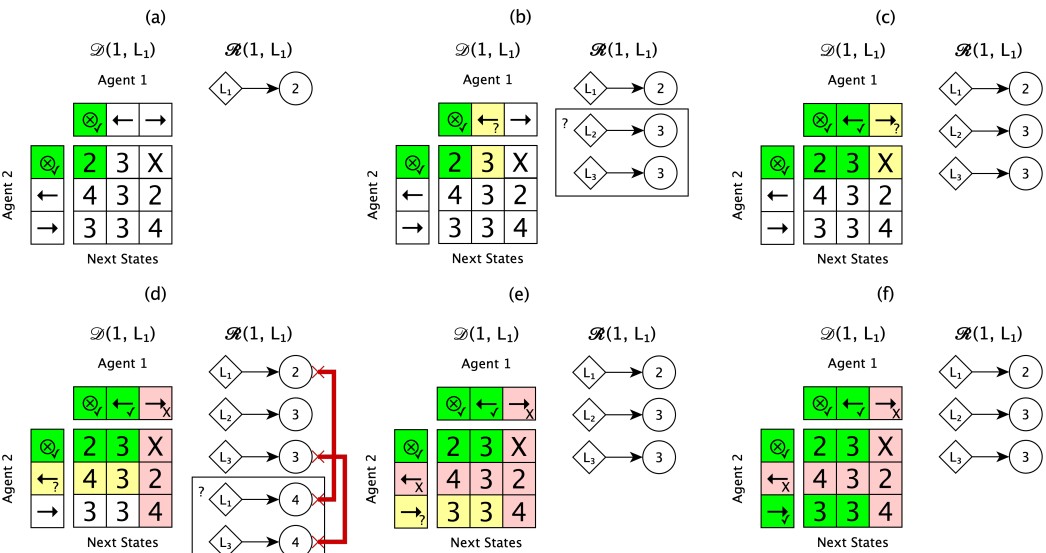

Figure 2: A detailed view of Algorithm 1, which finds a Cartesian set of safe, unambiguous actions. The shield from Figure 1 is used as input. This figure focuses on the process of finding safe actions for state 1, after $L_1$ has been observed. The 3x3 grid of squares represents the joint actions allowed by the centralized shield for a given state-label pair. The 3x1 rectangles on the top and left represent the individual actions which the decentralized shield will allow. Green with a check mark represents that the individual action is allowed, red with an "X" represents that the action is not allowed, and yellow with "?" represents that the action is under consideration. The mapping from diamonds to circles represents knowledge of what the next state will be, given the next shield. (a) A single safe joint action—$(\bigotimes, \bigotimes)$—is chosen arbitrarily. $\mathcal{R}$ reflects that if the agents see $L_1$ next, state 2 comes next. (b) The agent considers adding agent 1, action $\leftarrow$; it is safe so this action is allowed. (c) The algorithm considers agent 1, action $\rightarrow$. It may lead to an unsafe state, so this individual action is rejected. (d) The algorithm considers agent 2, action $\leftarrow$. This would add two joint actions. Adding state 4 as a possible next state means that the states are ambiguous if $L_1$ or $L_3$ is observed, so it is rejected. (e) The algorithm considers agent 2, action $\rightarrow$. Both joint actions which this would add are safe, so it is allowed. We don't consider the effects of action $(\rightarrow, \rightarrow)$, as agent 1 is prevented from selecting $\rightarrow$. (f) The finished result, used for the next step of the shield decentralization procedure.

No matter what, the decentralized shield which our algorithm generates will always allow at least one safe joint action, in a manner similar to the trivial solution described above. In practice (c.f. Section 6), we find that the resulting decentralized shield tends to allow a large range of joint actions, even on input shields which are not fully unambiguous or Cartesian.

## 6 Experiments

We aim to show with our experiments that, as predicted by Theorem 1, decentralized shields correctly prevent the agents from taking unsafe actions, while an equivalent unshielded agent may take unsafe actions in the same situation. Furthermore, we hypothesize that agents which use our decentralized shielding method perform comparably to agents which use a centralized shield.

### 6.1 The Gridworld Collision Domain

We adapt the gridworld maps from Melo and Veloso [16] to test our method. These maps, shown in Figure 3, have previously been used to test multiagent shielding, with an assumption of instantaneous local communication [10]. Each agent has five actions available: movement in the four cardinal directions, or a no-op. If all agents reach their goal positions, they each receive a +100 reward. Any agents which hit a wall receive a -10 reward. The safety specification is that a collision between two agents should never occur (including agents crossing over each other). In an unshielded

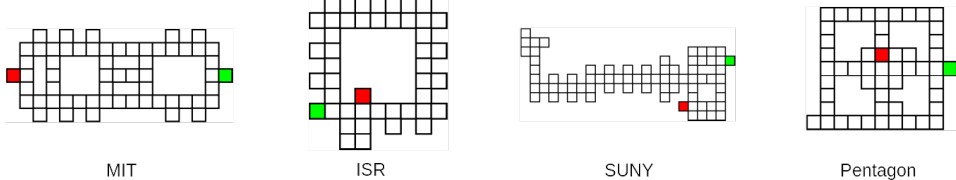

|  MIT | ISR | SUNY | Pentagon |

Figure 3: The four gridworld environments adapted from Melo and Veloso [16]. One agent each starts in the red and green squares; the goal is to switch positions.

environment, if an agent violates this specification, all agents receive a -30 reward. Agents receive a -1 reward at all other time steps. We use the four maps "MIT," "ISR," "SUNY," and "Pentagon," both with and without randomized start positions. These maps range in size from 8x10 to 10x23.

We synthesized a centralized shield which avoids collisions, using a reactive synthesis tool [9]. Note that this shield is shared across all four environments—it is a coarse abstraction which operates solely on relative agent positions, ensuring collision-free behavior in any gridworld with similar dynamics. The synthesis tool outputs a shield with many redundant states; we remove duplicate and unreachable states. In the resulting shield, each environment label only appears in a single state, so the resulting shield is trivially unambiguous (though it is not fully Cartesian).

We then utilized the procedure described in Section 5 to obtain a decentralized shield which works for all of our gridworld environments. The resulting individual shields are more conservative when the agents are near each other, ensuring that no more than one agent moves into a potentially contested area. Centralized shield synthesis took approximately five minutes on a M1 Macbook Pro, and our decentralization algorithm ran in under 30 seconds.

We analyzed the structure of the shields to determine how much the decentralization procedure restricts the set of actions available. With the centralized shield, there are an average of 23.30 safe joint actions available to the agents for a given shield state. Using the trivial solution from Section 5, every shield state would only have 1 joint action available. Our decentralization algorithm produces a shield where there are an average of 22.07 joint actions available—not much more restrictive than the centralized shield.

We trained independent tabular Q-learning agents using $\epsilon$-greedy exploration, with a linear $\epsilon$ annealing schedule from 1 to 0.05. The discount factor is 0.9. Agents were trained with a centralized shield, a decentralized shield, and with no shield. When an agent attempts to take an action $a$ which is not allowed by the shield, the penalty reward for the synthetic transition $r_p = -10$. These hyperparameters were chosen to closely match the parameters from [10], where possible.

Agents were trained for 2.5 million timesteps each, using 10 different random seeds for each configuration. Each individual run took approximately 5 minutes on a single core of an Intel server CPU; we trained 240 agents in total. This was sufficient for convergence in all variations. Results are shown with both fixed and random starting positions in Table 1. Extended tables and training curves are listed in the appendix.

As guaranteed by Theorem 1, this experiment shows that, because the centralized shield protects the agents from taking such unsafe actions, the agents which use a decentralized shield are also prevented from taking unsafe actions. The experiment also reveals that neither shielding method consistently performs better than the other method, as the highest-performing agent is generally within the margin of error. Therefore, our decentralized shielding approach allows safe, high-quality learning during training and execution in environments without communication. The training curves (Figure 4, in the appendix) tell the same story—no method works significantly faster during training.

## 6.2 The Relative Particle + Momentum Domain

One advantage of formally synthesizing a shield, rather than naively blocking actions which directly lead to an unsafe state, is that a shielded agent can avoid states which do not immediately violate a specification, but where all future paths end in an unsafe state. This can occur in domains with momentum; if a car is driving towards a cliff, it must slow down well before it reaches the edge. Therefore, we would like to evaluate agents which use our shield in momentum-based environments.

Table 1: Agent performance after 2.5 million environment steps, using the same shield for training and testing. Results show average evaluation return and standard error over 10 seeds, as well as average total violations over 100 testing episodes in parentheses. Note that centralized shielding is only for comparison; in a no-communication environment, this method would not be feasible.

| Start Type | Map Name | Centralized | Decentralized | No Shield |
|---|---|---|---|---|
| Fixed | ISR | $90.4 \pm 0.6$ (0) | $89.9 \pm 0.5$ (0) | $90.0 \pm 0.6$ (12.3) |
| | MIT | $72.0 \pm 0.8$ (0) | $71.6 \pm 0.3$ (0) | $72.8 \pm 0.5$ (0) |
| | Pentagon | $89.0 \pm 0.4$ (0) | $82.0 \pm 5.8$ (0) | $89.7 \pm 0.3$ (4.8) |
| | SUNY | $83.3 \pm 1.4$ (0) | $86.6 \pm 0.6$ (0) | $86.2 \pm 0.4$ (0) |
| Random | ISR | $78.6 \pm 2.2$ (0) | $76.1 \pm 2.0$ (0) | $83.6 \pm 1.4$ (2.0) |
| | MIT | $82.6 \pm 0.2$ (0) | $83.2 \pm 0.2$ (0) | $81.4 \pm 0.9$ (0.4) |
| | Pentagon | $88.4 \pm 0.5$ (0) | $80.3 \pm 3.5$ (0) | $89.1 \pm 0.3$ (0.9) |
| | SUNY | $78.1 \pm 0.4$ (0) | $73.1 \pm 1.7$ (0) | $77.3 \pm 0.4$ (0.4) |

Table 2: Individual DQN Agent Performance in the Particle-Momentum domain.

| Observability | Start Type | Centralized | Decentralized | No Shield |
|---|---|---|---|---|
| Partial | Fixed | $70.9 \pm 5.8$ (0) | $82.8 \pm 2.3$ (0) | $67.8 \pm 3.0$ (34.3) |
| | Random | $70.8 \pm 1.7$ (0) | $80.5 \pm 1.4$ (0) | $33.2 \pm 2.4$ (115.7) |
| Full | Fixed | $91.6 \pm 0.0$ (0) | $91.6 \pm 0.0$ (0) | $90.7 \pm 0.2$ (3.0) |
| | Random | $94.6 \pm 0.1$ (0) | $94.6 \pm 0.1$ (0) | $93.7 \pm 0.2$ (2.8) |

We introduce an environment as follows. There are two agents, representing particles on a discretized grid. Agent 1 begins 9 units above, and 9 units to the left of agent 2, with both agents at rest. Both agents obtain a reward of 100 when agent 1 gets to the position 9 units to the right of and below agent 2. If the agents get more than 10 units apart in any individual dimension, or collide or cross with each other, it is considered a safety violation (and the agents get a reward of -30). Otherwise, all agents get a reward of -1 at each time step. Each agent can move in any of the four cardinal directions, or do nothing. Additionally, agents have *momentum*; an agent's action is added to its previous movement to obtain its next movement in the environment. The relative velocity is capped at 2 units per time step in each direction. In the fully observable variant of this environment, both agents observe the relative positions and relative velocities. In the partially observable version, the agents only observe the relative positions.

We synthesized shields which only use the relative positions of the agents as the label set; such shields can be utilized in either variant of the environment. Using the same structure analysis as in Section 6.1, we calculated that the centralized shield allows an average of 21.67 actions, while the decentralized shield allows an average of 20.81 actions.

To show that our decentralized shielding algorithm is also effective regardless of the underlying reinforcement learning algorithm used by the agent, we trained reactive DQN agents to solve this task [17]. Full hyperparameters for this agent are located in the appendix. 10 random seeds were used in the fully observable case, and 50 seeds were used in the partially observable version of the environment. Each run took approximately 8 hours using 3 threads on a server CPU.

The results are shown in Table 2. Under full observability, we reach similar conclusions as in the gridworld-collision domain: the agents protected by centralized and decentralized shields both achieve comparable performance. As guaranteed by Theorem 1, the agent which uses the decentralized shield is as safe as the agent which uses a centralized shield. This is further emphasized under the partially observable variant of the environment, where the unshielded reactive agents do not have sufficient information about the environment, and thus behave in a wildly unsafe manner. In contrast, the shields are able to infer the current momentum based on the incremental changes in relative agent positions, leading them to never take an unsafe action.

## 7 Broader Impact and Limitations

We believe that the algorithms presented in this work represent progress towards the more general goal of safe reinforcement learning. Like most provably correct methods, shielding suffers from scalability issues such as the well-known state explosion problem [3]. For example, prior to de-duplication and removal of unreachable states, the shield which we synthesized for the gridworld experiments contained 71130 states. The particle environment's shield, including momentum information, contained 170348 states. The size of a centralized shield tends to scale exponentially with the number of agents. Our decentralization algorithm takes linear time with respect to the number of state-label pairs in the input shield, so we are well positioned to take advantage of methods which synthesize a centralized shield without such state explosion.

Additionally, obtaining an environment abstraction is difficult—we wrote several iterations of our abstractions, each containing subtle but serious bugs, before finally obtaining precise and accurate models of the environments. If our algorithm is given an inaccurate centralized shield, the decentralized shield which the algorithm outputs cannot be expected to perform well. We therefore caution against overreliance on an abstraction which may not accurately represent the environment.

Furthermore, there exist some environments where it is difficult, or even impossible, to determine a shared label set. For example, an environment with highly asymmetric observations would not satisfy the assumptions of the shielding framework we use. However, there may still be sufficient information to ensure safety in these challenging environments; we are working on developing methods to synthesize a decentralized shield without a shared label set.

As with any advance in machine learning, it may be possible to use our method to safely train an agent to perform an unethical task. However, we hope that as a whole, a process for safe reinforcement learning training and execution will reduce potential societal harms resulting from AI.

## 8 Conclusion & Future Work

Our experiments show that the shield decomposition method described in this paper results in shields that are safe and do not require any communication between agents. Furthermore, agents which are trained and evaluated with a decentralized shield perform comparably to agents which are trained and evaluated with a centralized shield, or without any shield.

We are currently working to extend our results in several ways. First, we are implementing additional agents and training schemes, including individual Deep Q Networks with convolutional layers [17], and Centralized-Critic Decentralized-Actor methods [11]. Second, we are investigating methods to scale our shield decomposition method to the more complicated shields and safety specifications necessary to describe complex continuous or partially-observable environments. Our algorithm makes a number of arbitrary choices which do not impact safety, but which could affect shield performance. It may be useful to define additional metrics for shield performance and permissiveness that do not necessarily involve training and testing agents over thousands of episodes, and to use these metrics to improve the decisions that our algorithm makes. We would like to eventually develop a method which obtains a maximally permissive decentralized shield, no matter what properties hold for the input shield. Finally, although similar problems in formal methods are undecidable [18, 21], we would like to explore heuristic methods for direct decentralized shield search and synthesis.

## Acknowledgments and Disclosure of Funding

We would like to thank Ingy ElSayed-Aly for her assistance with shield synthesis. We would also like to thank the reviewers for their thoughtful comments and suggestions.

This work has been supported by NSF SaTC Award CNS-1801546, NSF CAREER Award IIS-2044993, and Army Research Office Award W911NF2010265. This work was completed in part using the Discovery cluster, supported by Northeastern University's Research Computing team.

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
