# OpenReview forum: "Shield Decentralization for Safe Multi-Agent Reinforcement Learning"
_NeurIPS.cc/2022/Conference — NeurIPS 2022 Accept_

### Official Review · Reviewer_4UYh · 2022-06-30

**Rating:** 8
**Confidence:** 4
**Soundness:** 4 excellent
**Presentation:** 3 good
**Contribution:** 4 excellent

**Summary:**

In this paper, the authors address the problem of safe multi-agent decision making in decentralized communication free environment. They build upon previous approaches generating centralized safety shield in multi-agent problems to propose an algorithm that generates a decentralized safety shield given the centralized one.
The algorithm consists of a search procedure, it first identifies one safe actions for every agent at every states. Then it is progressively extending that set for each agent. To extend the set it checks that every new actions to be added still satisfies the safety constraints and is unambiguous. Some proofs are provided in appendix.

The authors empirically analyze the decentralized shielded generated by their algorithm on two discrete environments. In both cases the shield is generated to prevent collisions between two agents in a discrete grid world domain.


**Questions:**

1. What is the example problem in figure 1 and 2? What are the red and green square, what are the circles?
2. For future work, would it be more efficient to directly search for decentralized shield rather than computing a centralized shield first and then decentralizing it?
3. Could you clarify whether or not each agent need to observe the joint state?
4. What is the benefit of algorithm 2 and 3 over algorithm 1? This was not obvious from reading the paper.


**Limitations:**

The authors are upfront and clearly explained the limitation of their method and impact in section 7. They mention possible extension for future work. The most important one is to address the scalability issue. The authors are also very clear about experimental challenges on the design of the environment.

**Strengths And Weaknesses:**

The grade has been updated after clarification by the author on the figure illustrating how the proposed algorithm works.

The problem being addressed is relevant and important as multi-agent decision making and safe decision making can be applied to many societal applications. Such concept of decentralized shielding and the approach to generate the decentralized shield is to the best of my knowledge novel.

Overall the paper is really clear and well-written, the introduction, related work and problem setting are very nice to read. The related work section is quite clear and they clearly highlight how their method builds upon existing work in a non-trivial way by proposing an algorithm decentralizing an existing centralized shield. They do a very good job explaining a naïve approach to build such shield and how it can be improved. Some intuitive explanation is always provided to accompany the formalism and it is greatly appreciated.

The experiments are satisfactory, however some discussion is missing on how it would handle more complex safety specification and scale to more than 1 agents. It would have been very nice to introduce another problem that is not related to collision avoidance.

 The clarity of figure 1 and figure 2 is insufficient though. I read the caption multiple times and I still don’t get it. See questions. It almost seems like one paragraph is missing from the paper. Maybe the author could remove a figure and add theorem 1 which is referred to many times. Luckily algorithm 1 is quite ok to follow.

An aspect that was unclear to me is how the joint state is handled, it seems the authors assume that the joint state is fully observable by all the agent and there is one mention that it could be extended to individual state. It seems like such task is highly non trivial. The fact that the agent observe the joint state makes me question the initial claim of communication free environment.

---

> ### Author Response · Authors · 2022-07-29
> **Reply to Reviewer 4UYh**
>
> Thank you for your review.
>
> 1. On the question of the figures, Figure 1 was our attempt to provide a visual representation of the whole sequence of algorithms; the leftmost column as a representation of the output of Algorithm 1, the central column for Algorithm 2, and rightmost for Algorithm 3. In all three columns, the circles represent states in the shield, while arrows represent transitions to other states. Colors on the arrows represent labels which are observed from the environment, while the grids of green and red boxes represent actions which are allowed or disallowed. The shield which is depicted in the figures is not meant to enforce safety for any specific task; rather, it is a simplified shield used to demonstrate key parts of the sequence of algorithms.
>
>     The leftmost column of Figure 1 represents that from the initial state, after observing the label “brown” from the environment, there are two possible states which the centralized shield may transition to. The colored boxes indicate the safe actions which are discovered by Algorithm 1: Agent 1 may take actions 1 or 2, while Agent 2 may take any actions.
>
>     The central column represents that Algorithm 2 has projected the set of safe actions onto two transient-state individual shields; one shield per each agent. The small black circles represent transient states. Using Agent 1 (top) as the example, the initial state is a transient state where either labels brown or blue may be observed. After observing brown, the shield would transition to the upper state. From here, the previous step determined that Agent 1 may only take actions 1 or 2. No matter which of these allowed actions get taken, the next state is the transient state on the top-middle. After observing a label from the environment again, the shield would transition to a state following that.
>
>     The right column represents that Algorithm 3 has combined transient states with the states which follow, in order to obtain a DFA which matches the shield interface. From the initial state, if the agent observes the brown label, and takes allowed actions 1 or 2, it transitions to the upper state.
>
>     Figure 2 visualizes each step of Algorithm 1 in more detail. (a) represents the input to the algorithm. Note that this is a different centralized shield than the input represented in Figure 1. In (b)-(f), the 3x3 grid of colored squares represents the joint actions which are allowed by the centralized shield for a given state/label pair. The 3x1 rectangles on the top and left represent the individual actions which the decentralized shield will allow. Green represents an action which is allowed by the shield, red represents an action which is not allowed, and yellow indicates that a given individual action is under consideration during that step of the algorithm. The list of squares with arrows to circles represents $\mathcal{R}(s, l)$; the map of what labels of possible next shield states correspond to specific shield states; this is necessary to detect ambiguous actions (indicated by the same color pointing to two different states, such as in (e)).
>
>     The revised version of the paper will have an extra page which we can use to clarify the figure, its caption, and the text in the main paper. Additionally, we plan to update Figures 1 and 2 to use the same example shield as each other.
>
> 2. On the question of synthesizing a decentralized shield directly, rather than computing a centralized shield first: We do not yet have an algorithm to do so, but it is an interesting, if difficult, area to explore. A general algorithm may run into issues such as the undecidability of decentralized reactive synthesis, so the decomposition algorithm may still be more efficient than directly searching for a decentralized shield. Such approaches and the resulting analysis is left for future work.
>
> 3. On the question of the observations necessary to use a shield: Our work is based on the centralized shielding literature, which typically assumes an MMDP formalization of the environment; the state is fully-observable. However, our method continues to work in some partially-observable cases; for instance, in tasks such as Experiment 2, where all agents receive enough information to ensure safety, but not necessarily enough information to solve the task without memory. While our work is the first to eliminate the communication requirement for shielding, we leave the problem of shield synthesis with more constrained observations to future work.
>
> 4. On the question of Algorithm 1 versus Algorithms 2 and 3: the process for constructing a decentralized shield is the full sequence of Algorithms 1, 2, and 3. Algorithm 1 is the most important/novel step of the process, while Algorithms 2 and 3 transform the outputs of Algorithm 1 into DFAs which satisfy the standard shield interface. We will clarify these points in the revised paper.

---

> > ### Comment · Reviewer_4UYh · 2022-08-03
> > **The clarification reinforced the fact that the method is sound and useful**
> >
> > Thank you for your reply.
> > 1. Your explanation of the figure was extremely useful in better understanding the algorithm and it seems an easy revision to make to the paper. I am willing to update my review. I would suggest the following for the final version of the paper:
> > - add labels to each of the column (algorithm 1, 2 and 3)
> > - add labels to the grids (agent1 action vs agent 2 action)
> > It was not obvious that colors were labeled. If you could add a paragraph equivalent to your answer that would be even better.
> > 3. Ok, it would be good to make it explicit. I understand that theoretically it does not imply any communication when executing the policy. This is a bit outside of the scope of the paper but practically, it seems that if we want a fully observable state e.g. the robots know each other's position, communication would always be needed?
> > 4. Ok please add to the main paper.

---

> > > ### Author Response · Authors · 2022-08-05
> > > **Clarification about observability**
> > >
> > > Thank you for your review and recommendations; we will implement these in the final version of the paper.
> > >
> > > For observability: in some environments, agents may be able to independently reconstruct the safety-relevant information from cameras, sensors, etc. In environments where agents are completely blind and cannot communicate, it may be impossible to enforce safety, unless exact paths are planned in advance. The spectrum of environments between these extremes is quite interesting however, and we are interested in exploring safety in communication-free environments with local or imperfect observability in the future.

---

### Official Review · Reviewer_iC7K · 2022-07-10

**Rating:** 8
**Confidence:** 4
**Soundness:** 4 excellent
**Presentation:** 4 excellent
**Contribution:** 3 good

**Summary:**

This paper considers the problem of reinforcement learning in the scenario in which some states are unsafe, and an agent cannot be allowed to visit them, not even as part of exploration. An example of such a scenario would be a robot behaving in a physical environment, in which it should not collide pedestrians, who might be hurt from such a collision. Due to this safety-critical nature of the problem, it is not sufficient to assign a negative reward to those states and expect the agent to learn to avoid those states from experience.

Existing work has considered the task of learning "shields", which are a state-dependent subset of actions that ensure that the agent(s) will never visit the unsafe states. It is possible to generalize the notion of shields to the multi-agent scenario by requiring that the joint actions taken by all of the agents satisfy the constraints of the (centralized) shield, but this has the drawback that it requires the agents to communicate and coordinate their actions, as well as for the shield to have enough information about all of the agent behaviors to know which state restrictions it should impose next.

This work considers the problem of automatically decomposing a centralized shield into a set of decentralized shields, one for each agent. The shields are iteratively constructed by adding actions for agent that do not result in a violation of the centralized shield. The authors demonstrate their approach on a collection of RL case studies.

**Questions:**

- If the centralized shield is Cartesian and unambiguous, the authors claim in line 172 that their algorithm will produce maximally permissive shields. This statement is given casually, and does not even receive an informal argument to justify it. This statement further seems to be contradicted on line 325. Does this procedure produce maximally permissive decentralized shields, even in the restricted case of cartesian and unambiguous shields?

- For a given centralized shield, is the decomposition unique? This is not clear to me, it is not stated in the paper, and if a decomposition is not unique, it may be valuable to be able to be able to search among decompositions to balance application-specific trade-offs.

- Is it possible to derive a result on a bound for how conservative a distributed shield will be? It seems that for the case of finite MMDPs, it should be possible to quantify how many behaviors are allowed by the centralized shield and disallowed by the decomposition. It would be useful for practitioners to be able to quantify how much of a penalty they are paying in a specific application.

- The authors mention informally that the complexity of computing a shield decomposition is linear in the size of the centralized shield--but how does this relate back to the complexity of the original MMDP? I expect that the complexity of computing a centralized shield is known, and I think the paper would be improved by expressing the complexity of a distributed shield from the size of the MMDP.

- The authors seem to be familiar with the formal methods literature. Existing work in that literature considers the problem of decomposing safety properties into contracts for components in a large architecture--how does their work relate to the contract decomposition literature?

**Limitations:**

I think the limitations were suitably addressed.

**Strengths And Weaknesses:**

- The work is certainly relevant. Distributed safe reinforcement learning is an important topic with a broad array of useful applications.

- The case studies demonstrate feasibility of the approach.

- The soundness theorem seems to be correct.

- The metric on permissiveness is given in terms of average actions per state, but this may be the wrong metric. In some cases, a few actions in a specific state may disallow a large portion of the behavior-space, resulting in excessively conservative behavior that is not accurately captured by the metric considered. For finite MMDPs, it should be possible to quantify the percentage of the behavior-space that is lost, and I think this would make for a better metric.

- I have a few open questions, some of which may be straightforward for the authors to address in the camera-ready version.

---

> ### Author Response · Authors · 2022-07-29
> **Reply to Reviewer iC7K**
>
> Thank you for your review.
>
> Please note that in this reply, we refer to Algorithm line numbers (starting from 1 on the line containing “Input”) which we mistakenly did not enable for the original submission; this will be fixed in the revised version.
>
> 1. On the question of permissiveness: the comment on line 325 refers to the general case, in which we currently do not know if the decentralized shield is necessarily maximally permissive. Future work can analyze this question further, and potentially improve the decentralized shield. The claim on line 172 refers to the specific case of shields which are Cartesian and unambiguous. We will make this clear in the revised version. We provide intuition below about why this type of decentralized shield is maximally permissive, and will add a more formal version of this argument to the revised paper:
>
>     Given a shield state and label, Algorithm 1 determines the set of individual actions the agents can take. $\mathcal{C}(s, l)$ represents the set of joint actions allowed by the centralized shield at this point. If the shield is Cartesian, at the `if` check on line 15, $A$ will always either be a subset of $\mathcal{C}(s, l)$, or fully disjoint; if the shield is unambiguous as well, there will be no joint actions which are allowed by the centralized shield, but then subsequently aren’t added to the decentralized shield on line 16. Since the decentralized shield allows every joint action which is present in the centralized shield, it is necessarily maximally permissive.
>
> 2. On the question of unique decomposition: the results are not necessarily unique. In particular, our algorithm allows for two arbitrary choices: First, for each state-label pair, the guaranteed-safe action $a$ is chosen arbitrarily from the set of safe actions allowed by the centralized shield (line 9 of algorithm). The no-op action, when safe, appears to be the most reasonable choice for this, although a full exploration of how to choose the starting point for the algorithm is left for future work.
>
>     Second, the order of agent priority may be chosen arbitrarily, and given as iteration order on line 12. Our current implementation randomizes the agent priority each time step using a deterministic PRNG, as hinted at by the comment on line 12; we found this to be a good trade-off between complexity and coverage of the joint action space.
>
> 3. On the question of quantifying a bound for how conservative a shield is: we did not extensively explore this idea, although the agent-priority randomization mentioned above is an attempt to allow as many behaviors as possible. It is an interesting question, and the development of additional metrics to quantify permissiveness is future work.
>
> 4. On the question of algorithmic complexity: as calculated in our response to Reviewer ZmUF, the time complexity of decentralizing a centralized shield is $O(S * L^2 * D * A * A^D)$, where $S$ is the number of states in the centralized shield, $L$ is the size of the label space, $D$ is the number of agents, $A$ is the number of actions per agent, and $A^D$ is the size of the joint action space.
>
>     In our experiments, the size of the centralized shield tended to be linear to the size of the state space of the MMDP; due to the specific implementation of the synthesis tool we used, there was an additional scaling factor of the number of joint actions available. It is worth noting that the MMDP's state space usually scales exponentially with the number of agents.
>
>     However, the exact scaling factor for the shield is domain-specific. For example, if we were to add complexity to the reward structure of our gridworld domain, but maintain the same collision-avoidance safety specification, we could likely use the same shield without further modification.
>
> 5. On the question of contract decomposition literature: we are generally familiar with compositional methods in verification, and with contract based design, and we will add references to these methods in our revised paper. We are not specifically aware of the contract decomposition literature that the Reviewer is referring to, and we would be grateful if the Reviewer could point us to specific papers in that area. Thank you.

---

### Official Review · Reviewer_ZmUf · 2022-07-16

**Rating:** 5
**Confidence:** 4
**Soundness:** 3 good
**Presentation:** 3 good
**Contribution:** 2 fair

**Summary:**

The authors present a method to decompose a centralized shield in a multi-agent system to individual shields for each agents that can enforce a safety specification with no communication between agents. The authors' proposed approach takes a centralized shield and outputs a maximally permissive decentralized shield. The authors then show, using experiments on gridworlds, that their decentralized shields can perform comparably to a centralized shield and significantly outperforms no shield at all.

**Questions:**

Can the authors provide any theoretical complexity results on their decomposition process? It is stated simply in section 7 takes linear time in the number of state-label pairs of the centralized shield. A proof, or at least intuition for this statement will be helpful.



Is there a proof the resulting decentralized shield is indeed maximally permissive?

**Limitations:**

Adequately addressed.

**Strengths And Weaknesses:**

This paper, while writen clearly, needs significantly more detail and preciseness. The paper also needs better structure as it is not easy to find out what exactly the problem statement is and what the inputs to the problem are and the outputs of the solution. Clearly stating the problem statement in the paper would be helpful.

The focus of this paper is on the no-communication aspect of the multi-agent shielding problem. However, one of the main issues with multi-agent shielding is that centralized synthesis often scales very poorly. This aspect is not addressed much by the authors, which can be problematic as their solution requires the input of a centralized shield which is often not computationally feasible to procure.

Furthermore, the paper would benefit greatly from complexity results. One of the key contributions is the decomposition of the centralized shields, but there is no intuition on the scaleability of the process. Additionally, it could also help to report synthesis times in the experiments.

---

> ### Author Response · Authors · 2022-07-29
> **Reply to Reviewer ZmUF**
>
> Thank you for your review.
>
> The key problem which our paper addresses is stated on line 162 (Problem 3), although by necessity, some detail was left out of the paper due to the length limitation. We plan to use part of the extra page allowed in the camera-ready version of the paper to enhance the clarity of the problem statement and algorithm.
>
> Please note that in this reply, we refer to Algorithm line numbers (starting from 1 on the line containing “Input”) which we mistakenly did not enable for the original submission; this will be fixed in the revised version.
>
> 1. On the question of scaling and algorithmic complexity, our paper provides a solution to a key limitation in current state-of-the-art methods (such as ElSayed-Aly et al.)—the need for communication with a centralized shield during execution. Our approach does assume a centralized shield is given, but generating a centralized shield is a common task, and there are several tools to do so.  As a result, our method is widely applicable, and future work can focus on removing or scaling the centralized shield synthesis problem. We note that, in our approach, the centralized shield is only required during the decentralized shield generation process, and not during the decentralized shield execution.
>
>     Here is an informal proof that the time complexity of Algorithm 1 is linear in relation to the size of the shield; we will add a version of the following to the appendix of the revised paper:
>
>     ```
>     S = number of shield states
>     A = number of actions per agent
>     L = number of labels
>     D = number of agents
>     A^D = size of the joint action space
>
>     Line 7: Loop, iterate O(S * L) times
>       Line 8: O(A^D)
>       Line 9: O(A^D)
>       Line 10: O(D)
>       Line 11: O(1)
>       Lines 12-20: Loop, iterate D times:
>          Lines 13-19: Loop, iterate A times:
>             Line 14: O(A^D)
>             Line 15: O(L * A^D)  # Note: this has a small coefficient if `UnambiguousActions` is implemented efficiently
>               Line 16: O(1)
>               Line 17: O(A^D)
>     ```
>
>     The time complexity is $O(S * L^2 * D * A * A^D)$. Note that $L$, $D$, and $A$ are typically small constants, independent of the shield size.
>
>     On a laptop with a M1 Max processor, the shield for experiment 1 took 273 seconds to synthesize, and the shield for experiment 2 took 326 seconds. The vast majority of this time was spent in centralized shield generation; our decentralization step took only a few seconds. We did not attempt to optimize the centralized shield synthesis and such optimization may (significantly) reduce the runtime. We will add these numbers to the appendix.
>
> 2. On the question of permissiveness: for the general case, we do not know if the decentralized shield is necessarily maximally permissive. A full analysis of this question is left for future work. For the specific case of shields which are Cartesian and unambiguous, we provide intuition below about why this type of decentralized shield is maximally permissive, and will add a more formal version of this argument to the revised paper:
>
>     Given a shield state and label, Algorithm 1 determines the set of individual actions the agents can take. $\mathcal{C}(s, l)$ represents the set of joint actions allowed by the centralized shield at this point. If the shield is Cartesian, at the `if` check on line 15, $A$ will always either be a subset of $\mathcal{C}(s, l)$, or fully disjoint; if the shield is unambiguous as well, there will be no joint actions which are allowed by the centralized shield, but then subsequently aren’t added to the decentralized shield on line 16. Since the decentralized shield allows every joint action which is present in the centralized shield, it is necessarily maximally permissive.

---

> > ### Comment · Reviewer_ZmUf · 2022-08-07
> > **Permissiveness clarification was helpful**
> >
> > I appreciate the comment regarding the decentralized shield permissiveness. While I do think it is an important aspect, providing an informal intuition is very helpful.
> >
> > My other concern was regarding providing centralized shields. I agree that creating a shield is easy and there exists many tools do so, my concern was the computational intractability of constructing a centralized shield for a multi-agent system. The work in this paper assumes that there exists such a centralized shield which is in practice is usually not computationally feasible and hence I think this assumption is a very strong one.
> >
> > In saying that however, I do believe the rest of the theoretical contributions of this paper are valid and interesting. I will bring my score up to a 5.

---

> > > ### Author Response · Authors · 2022-08-08
> > > **Reply to “Permissiveness clarification was helpful”**
> > >
> > > Thank you for your comments. We will add a note about the complexity of multi-agent centralized shield synthesis to the “Limitations” section.

---

### Meta-Review · Area_Chair_A6Dk · 2022-08-24

**Recommendation:** Accept
**Confidence:** Certain

**Metareview:**

It is agreed among reviewers that the paper should be accepted. Hope the authors can address the comments from the reviewers in the final version as promised.

**Award:**

No

---

### Decision · Program_Chairs · 2022-09-14

Accept